# Temperature Induced Gelation and Antimicrobial Properties of Pluronic F127 Based Systems

**DOI:** 10.3390/polym15020355

**Published:** 2023-01-10

**Authors:** Alexandra Lupu, Irina Rosca, Vasile Robert Gradinaru, Maria Bercea

**Affiliations:** 1“Petru Poni” Institute of Macromolecular Chemistry, 41-A Grigore Ghica Voda Alley, 700487 Iasi, Romania; 2Faculty of Chemistry, Alexandru Ioan Cuza University of Iasi, 11 Carol I Bd., 700506 Iasi, Romania

**Keywords:** thermosensitive gels, Pluronic F127, polysaccharide, curcumin, viscoelastic properties, antimicrobial activity

## Abstract

Different formulations containing Pluronic F127 and polysaccharides (chitosan, sodium alginate, gellan gum, and κ-carrageenan) were investigated as potential injectable gels that behave as free-flowing liquid with reduced viscosity at low temperatures and displayed solid-like properties at 37 °C. In addition, ZnO nanoparticles, lysozyme, or curcumin were added for testing the antimicrobial properties of the thermal-sensitive gels. Rheological investigations evidenced small changes in transition temperature and kinetics of gelation at 37 °C in presence of polysaccharides. However, the gel formation is very delayed in the presence of curcumin. The antimicrobial properties of Pluronic F127 gels are very modest even by adding chitosan, lysozyme, or ZnO nanoparticles. A remarkable enhancement of antimicrobial activity was observed in the presence of curcumin. Chitosan addition to Pluronic/curcumin systems improves their viscoelasticity, antimicrobial activity, and stability in time. The balance between viscoelastic and antimicrobial characteristics needs to be considered in the formulation of Pluronic F127 gels suitable for biomedical and pharmaceutical applications.

## 1. Introduction

Last decades, stimuli-responsive polymers were extensively investigated due to their high potential for many applications. Among them, the self-assembling of amphiphilic block copolymers into micelles and the temperature-induced gelation present a high interest in biomedical and pharmaceutical applications, as carriers for drug delivery [1,2,3], in regenerative medicine or dentistry [4,5,6], for preparing injectable hydrogels [7], in bioprinting [8], as sacrificial molding agents [9] etc. On the other hand, the proteins and renewable polysaccharides are non-toxic and endow some biological performances to biomaterials, such as biodegradability, biocompatibility, and mucoadhesive behavior. Thus, various polymers and biologically active compounds were used to develop suitable platforms for tissue engineering and regenerative medicine. Antimicrobial activity always represents a challenge for the design of new biomaterials because in some cases the enhancement of physicochemical performances implies a worsening of the biological properties or a weakening of the network structure. For example, the incorporation of different antimicrobial agents, such as curcumin and zinc oxide nanoparticles into the electrospun chitosan/polycaprolactone inhibited bacterial growth in the case of *Escherichia coli* and *Staphylococcus aureus*. However, as a decrease in cell viability was observed, curcumin inhibited the proliferation and survival of L929 cells in a dose-dependent manner [10].

Pluronic^®^ F127 (here abbreviated as PL) is a triblock copolymer approved by the US Food and Drug Administration for pharmaceutical and biomedical use. In its structure presents a central hydrophobic poly(propylene oxide) (PPO) block and two hydrophilic poly(ethylene oxide) (PEO) blocks, i.e., (PEO)_x_-*b*-(PPO)_y_-*b*-(PEO)_x_, with x = 100, y = 65. Due to the gel formation ability, PL is a versatile synthetic macromolecule used to prepare injectable hydrogels, cosmetics, or pharmaceutical materials. In aqueous solutions at low temperatures, PL solutions behave as low-viscosity fluids when unimers coexist with small aggregates. Above the critical micelle concentration (cmc), as temperature increases, the copolymer undergoes self-assembling forming a micellar structure, that spontaneously organizes into a shear-thinning, thermoreversible network [4,6,8,11]. The micellization and gelation processes, as well as the shape adopted by PL micelles (globular, cubic, or hexagonal), depend on thermodynamic conditions (solvent, temperature) and they are influenced by the addition of other pharmaceutical excipients or drug molecules [1,12,13,14,15,16].

The Pluronic-based gels present good printing properties and shape fidelity in physiological conditions, but they have modest cell viability [8] and structural integrity [17,18]. Some properties were improved in composites of PL with other synthetic polymers, such as poly(vinyl alcohol) [19], poly(aspartic acid) [20], poly(ethylene glycol) [21], or pharmaceutical ingredients [3]. The use of materials that combine the properties of PL with those of biomolecules is of high interest for bio-applications [15,22,23,24,25]. Usually, polysaccharides, proteins, or peptides are used for the design of composite hydrogels in order to enhance biological performances, including antimicrobial properties [26,27,28,29,30], to increase gel strength and bioadhesive properties [31], or these biomolecules are added to improve the residence time of the active compounds in tissues [15,24]. In the presence of polysaccharides, the morphological analysis revealed a denser structure with smaller pores as compared with PL pristine gels [31].

Dynamic light scattering investigations evidenced the coexistence of complex states of aggregation of Pluronic F127 aqueous solutions in the presence of chitosan or clay [32]. During the temperature-induced gelation, Pluronic has shown a fast and a slow diffusion process: the faster diffusion was associated with the interchange of unimers between copolymer micelles and, as the temperature increases, a slower diffusion occurs due to the formation of micellar clusters that undergo dehydration of the PO block and gelation. Entities of different sizes, such as large micellar aggregates formed during the early stage of micellization, determine a slow relaxation and delays in the sol-gel transition. Around the gelation temperature, the micelles organize into close-packed structures, thus reducing progressively the size of the diffusing entities and above the gelation temperature, the size of micelles remains constant [32]. The morphology analysis of Pluronic gels (carried out by scanning or transmission electron microscopy) showed that the structural changes in presence of N,N,N-trimethyl chitosan which hindered the close packing of PF127 layers, leaving more pores on the network surface that enhances the drug release. The micelles’ sizes vary from 50 nm to 200 nm in water/ethanol 90/10 (vol.) mixtures [33]. The presence of gallic acid induces antimicrobial activity in Pluronic hydrogels [34,35]. Thus, gallic acid was loaded into a dual-responsive (temperature/pH) hydrogel that contains Pluronic F-127, N,N,N-trimethyl chitosan and polyethylene glycosylated hyaluronic acid and used for the treatment of atopic dermatitis [34]. Pluronic F127/chlorhexidine nanoparticles encapsulated into chitosan methacrylate-gallic form a hydrogel with enhanced antimicrobial properties against *Staphylococcus aureus* (*S. aureus*) and *Escherichia coli* (*E. coli*) (>99.9%) [35]. This hydrogel presents well-formed Pluronic micelles and interconnected networks made by polysaccharide derivatives.

In physiological conditions, the gel strength increased for a mixture of 14% PL and alginate, and the gelation temperature became lower as the concentration of polysaccharides increased [36]. In addition, polysaccharides can act as stabilizers for Pluronic gels [17]. Silk fibroin was used to improve the mechanical properties and stability of PL gels by inter-micellar packing and physical crosslinking [37]. The interactions between the PEO and PPO blocks and other polymers or small molecules can alter physicochemical properties, including gelation ability [15,38]. For example, pullulan incorporation in PL gels is limited by the extended immiscibility gap in an aqueous solution [39]. Hydrophobic modification of polysaccharides with long hydrocarbon chains (higher than C16) increases the gel stability for more than 6 months [17].

The thermally induced gelation of PEO-*b*-PPO-*b*-PEO copolymers was extensively discussed in the literature. However, the selection of an appropriate injectable system that satisfactorily combines the viscoelastic and biological properties is quite difficult. In this context, the present paper is focused on the viscoelastic behavior of temperature-responsive gels based on Pluronic F127 in presence of four polysaccharides, i.e., chitosan, sodium alginate, gellan gum and k-carrageenan, a protein (lysozyme) and ZnO nanoparticles. The possibility of improving the antimicrobial properties of gel composites is explored and discussed.

## 2. Materials and Methods

### 2.1. Materials

The polymer samples—Pluronic^®^ F127 (denoted PL), chitosan low viscosity (CS), sodium alginate (Alg), gellan gum (Gell), kappa-carrageenan (κ-Carr)—and ZnO nanoparticles (<100 nm) were purchased from Sigma-Aldrich Co. (Taufkirchen, Germany). Curcumin (≥90%) and lysozyme were purchased from Carl Roth (Karlsruhe, Germany). All samples were used as received.

Stock solutions of PL (20% wt.) and polysaccharide (1% wt.) were prepared by mixing the polymer with sterilized ultrapure water (using an Ultrapure water system TKA GenPure UF/UV 08.2204, Thermo Scientific, High Wycombe, UK), with the exception of chitosan which was dissolved in 1% acetic acid solution. Polysaccharide solutions were freshly prepared at room temperature using a magnetic stirring system and then they were stored for 24 h at 4 °C, apart from Gell sample, which was left at room temperature. Due to the hydrophobic/hydrophilic polyether structure, PL is soluble in cold water and less soluble in warm water. Thus, PL solution was prepared at low temperature (by using an ice-water bath) and then left for 24 h at 4 °C. The biomolecules or nanoparticles were added to PL solutions and the final concentration of polymer (PL and polysaccharide) in all samples was 16.83% wt. The samples containing curcumin were prepared and stored in the dark. A poor photostability of curcumin in presence of organic solvents was earlier reported (half-life is lower than 15 min) [40,41].

The structure of the used chemicals is presented in Figure 1 and the composition of PL-based formulations is given in Table 1.

### 2.2. Methods

#### 2.2.1. Rheological Measurements

Rheological measurements were carried out by using an MCR 302 Anton-Paar rheometer equipped with Peltier device for temperature control and plane-plane geometry (upper plate diameter of 50 mm, gap of 0.5 mm). During the experimental investigation, a solvent trap was used to generate a humidified environment in the vicinity of the sample.

The shear viscosity (η) of the samples was determined as a function of shear rate in stationary shear flow conditions, for shear rates ($\dot{\gamma }$) in the range 0.001–1000 s^−1^.

In oscillatory shear conditions, the elastic (G′) and viscous (G″) moduli were determined and they give information concerning the stored and dissipated energy during one cycle of deformation. The loss tangent (tanδ = G″/G′) is correlated with the degree of viscoelasticity of the sample (tanδ > 1 for viscous fluids and tanδ < 1 for elastic solids).

For PL-based samples, the sol-gel transition was evidenced through the evolution of viscoelastic parameters as a function of temperature from 5 °C to 80 °C, for a heating rate of 1 °C/min, at constant oscillation frequency (ω) of 1 rad/s and strain amplitude (γ) of 1%. The gelation kinetics were followed at 37 °C by using the samples stored in the refrigerator and poured on the lower plate of the rheometer thermostated at 5 °C. The temperature was then set at 37 °C and the viscoelastic behavior was followed in time (ω = 1 rad/s and γ = 1%). For samples thermostated 24 h at 37 °C, small-amplitude oscillatory shear experiments were carried out for oscillation frequencies between 0.1 rad/s and 100 rad/s, in the linear range of viscoelasticity (which was determined for each sample in amplitude sweep test).

#### 2.2.2. Antimicrobial Activity

The antimicrobial activity of the samples was determined by disk diffusion assay [42,43] against seven different reference strains: *Staphylococcus aureus* ATCC25923 (*S. aureus*), *Escherichia coli* ATCC25922 (*E. coli*), *Enterococcus faecalis* ATCC29212 (*E. faecalis*), *Klebsiella pneumoniae* ATCC10031 (*K. pneumoniae*), *Pseudomonas aeruginosa* ATCC27583 (*P. aeruginosa*), *Candida albicans* ATCC90028 (*C. albicans*), and *Candida glabrata* ATCC15126 (*C. glabrata*).

The microorganisms were stored at −80 °C in 20% glycerol. The bacterial strains were refreshed on nutrient agar (NA) at 37 °C. The yeast strains were refreshed on Sabouraud dextrose agar (SDA) at 37 °C. Microbial suspensions were prepared with these cultures in sterile solution to obtain turbidity optically comparable to that of 0.5 McFarland standards. Volumes of 0.1 mL from each inoculum were spread on Petri dishes. Then, the sterilized paper disks (6 mm) were placed on the plates and aliquots (50 μL) of the samples were added. To evaluate the antimicrobial properties, the growth inhibition was measured under standard conditions after 24 h of incubation at 37 °C.

To ensure the accuracy of the results, all tests were performed in triplicate. After incubation, the plates were analyzed using a scanning system SCAN1200^®^, version 8.6.10.0 (Interscience, Saint-Nom-la-Bretèche, France), and the results were expressed as the mean ± standard deviation (SD) performed with XLSTAT Ecology version 2019.4.1 software, Addinsoft, Paris, France [44].

## 3. Results

### 3.1. Rheological Behavior

#### 3.1.1. Rheological Behavior of Polysaccharides in Solution

Figure 2a shows the flow behavior of the four selected polysaccharide solutions in steady shear conditions at increasing and decreasing shear rates (37 °C).

All solutions exhibit Newtonian behavior at low shear rates and they become non-Newtonian fluids as the shear rate increases above a critical value ${\dot{\gamma }}_{c}$ when the viscosity starts to decrease. The longest relaxation time, τ, was determined as 1/${\dot{\gamma }}_{c}$, and its values are: 0.0488 s, 0.0976 s, 0.0318 s, and 0.0446 s for samples 2, 3, 4, and 5, respectively. Alg solution presents the highest viscosity and it exhibits pronounced thixotropy. At low shear rates, the viscosity of CS solution is close to those of κ-Carr solution and exhibits a lower thixotropy as compared with Alg solution. For the κ-Carr solution, the viscosity decrease is more pronounced at high shear rates ($\dot{\gamma }$ > 100 s^−1^) as compared with sample 1 (similar to Alg solution), and sample 4 did not present thixotropy.

In oscillatory shear conditions, only κ-Carr solution behaves as a Maxwellian fluid with G′~ω^2^ and G″~ω. CS, Alg and Gell solutions behave as structured fluids, with G′~ω^1.25–1.36^ and G″~ω^0.80–0.86^ (Figure 2b).

#### 3.1.2. Temperature-Induced Gelation for Various PL-Based Gels

Figure 3 presents the evolution of the complex viscosity as a function of temperature for all PL-based gels. At low temperatures, the viscosity of PL solutions increases 2–9 times in the presence of ZnO nanoparticles, lysozyme, Alg, or CS and with more than one decade by the addition of Gell or κ-Carr.

It was established that the micellization/gelation of PL in an aqueous environment is an endothermic process, entropy-driven [11]. By increasing the temperature, micelles with a hydrophobic core of PPO surrounded by a hydrophilic PEO shell are formed by expelling water molecules from the vicinity of hydrophobic blocks due to a strong diminishing of hydrogen bonds between water and PPO. As is observed in Figure 3a, the sol-gel transition of PL is shifted from 22 °C to lower temperatures by using biomolecules or nanoparticles (the highest effect is 7 °C given by ZnO nanoparticles). Only the presence of CS determines an increase in the transition temperature at 1 °C. At 37 °C, the gel state was reached for PL (sample 0) as well as for samples 5 to 10.

The viscosity and temperature profiles change considerably in the presence of curcumin (Figure 3b). Curcumin is a diarylheptanoid having an extended conjugated double bond structure (Figure 1), used as photosensitizer in topical photodynamic therapy [41]. Many studies have demonstrated the pharmacological potential of curcumin due to its biological, therapeutic, anticancer, and antitumor properties [45,46,47,48]. The first limitation of its application is its very low water solubility. For acid and neutral pH domains, it is slightly soluble (<50 nM); for basic pH values, the solubility increases but also a rapid hydrolytic degradation occurs [40]. Curcumin becomes soluble in the presence of cosolvents, such as ethanol or dimethyl sulfoxide [49]. However, the addition of organic polar solvents, salts, or urea in PL solutions influences the solvent quality and thus the micellization process [1]. We observed that the presence of ethanol in concentrated PL solutions prevents the micellization process, even at high temperatures. Kwon et al. [38] have observed that lower alcohols (methanol or ethanol) inhibit the micellization and gelation of PL, acting as “water-structure-breakers”, while higher alcohols (as for example butanol) act as “water structure-makers” and favor copolymer self-assembling in aqueous solution. On the other hand, it was reported that high concentrations of Pluronics completely monomerize curcumin and improve its dissolution through hydrophobic interactions formed in the core of the PPO micelles [41]. Curcumin resides in the hydrophobic core of PL micelles (it was previously shown that the size of the PPO core increases in the presence of curcumin [50,51]), and the hydrophobic interactions increase progressively over a larger domain of temperature (from 37 °C to 59 °C for sample 11 and 25 °C to 59 °C for sample 12, Figure 4c,d) as compared with sample 5 or sample 0 (from 23 °C to 26 °C for sample 0 that contain 0.33% acetic acid or from 24.5 °C to 31.5 °C for sample 5, Figure 4a,b). In addition, the acidic conditions (samples 5, 10, 11, and 12 contain 0.33% acetic acid) contribute to a delay of gelation. Sample 12 (containing curcumin loaded into PL/CS mixture) presents a gradual sol-gel transition that starts below 37 °C (Figure 4d).

#### 3.1.3. Gelation Kinetics at 37 °C

The kinetics of gelation was followed at 37 °C by using the samples stored in a refrigerator and poured on the lower plate of the rheometer thermostated at 5 °C. The temperature was then switched at 37 °C and the viscoelastic behavior was followed in time. For samples 6–10, the gelation profile is very close to the pure PL sample. The gel formation is very fast, occurring in the first 3–5 min after the temperature was switched from 5 °C to 37 °C. A different behavior appears only for CS (sample 5), when the sol-gel transition is also very fast, but the equilibrium structure needs more time to be reached. In the presence of curcumin, the dynamics of sol-gel transitions are slower, the manifestation of hydrophobic interactions is delayed and the network formation evolves in about 90 min (samples 11 and 12, Figure 5). The gelation time is an important characteristic for injectable gels; a very fast gelation at 37 °C is not preferred, the materials need a few minutes to be manipulated before reaching the desired place and being transformed into the desired shape.

In the present study, small amount of acetic acid improves the curcumin dissolution and its incorporation into the PL gels, whereas the presence of chitosan chains increases the particles’ stability and avoids the phase separation as observed previously in PL/curcumin aqueous systems [41]. The entire system, which comprised water and acetic acid (as the solvent), curcumin, PL, and polysaccharide, moves towards a less organized heterogeneous structure that influences the kinetics as the temperature rises (Figure 4d) or in isothermal conditions, at 37 °C (Figure 5b).

A similar system, containing curcumin dissolved in ethanol and incorporated into hydrogels containing a mixture of 23% Pluronics (F68 and F127) and up to 0.2% xanthan gum, was reported by Lee et al. [31]. Using this hydrogel instead of the curcumin solution in ethanol resulted in a significant increase in the total amount of curcumin that penetrated during transdermal delivery. The gelation occurs around 29.3 °C and G′ values at 37 °C are close to those obtained in the present study.

#### 3.1.4. Viscoelastic Behavior of Gels at 37 °C

Figure 6 shows the behavior of samples 0, 5, 11, and 12 in a solution of 0.33% acetic acid in amplitude and frequency sweep tests. The linear viscoelastic regime was reached for deformations γ ≤ 2 %. The elastic modulus increases in the presence of curcumin (samples 11 and 12) as compared with PL gel (G′ ≈ 1.2 × 10^4^ Pa).

A synergetic behavior is observed by chitosan addition (sample 12) for which G′ is about 6 × 10^4^ Pa and tanδ ≈ 0.06. After 24 h at 37 °C, the sample reached equilibrium, and the gel strength and stability increased. This suggests that curcumin increases the hydrophobic interactions via its hydrophobic aliphatic and aromatic moieties, and chitosan chains enhance the stability of the network through hydrogen bonds. In the sol state, random polymer chains and curcumin molecules are well dispersed due to hydrophobic interactions with PPO from PL. By increasing the temperature, the PL micelles form a network. When the organic solvent is added, the micelles start to diffuse into the solvent environment and delay or cancel the temperature-induced gelation process. In the presence of a polysaccharide, the PL micelles are surrounded by polymer chains and the composite gels are stabilized against their dissolution.

#### 3.1.5. Flow Behavior of Gels at 37 °C

PL-based gels are suitable as injectable materials in therapeutic applications [50,52]. The shear-thinning behavior is an important characteristic of such materials, as the viscosity decreases and the gel sample easily flows through the needle of the syringe when high shear forces are developed [53].

Continuous flow investigations carried out in stationary shear conditions revealed that all PL-based gels present injectability, and the viscosity decreases when the shear rate is above 0.01 s^−1^ (Figure 7). In the non-Newtonian domain, the shear viscosity (η) scales as ${\dot{\gamma }}^{n}$; the values of the power law index, *n*, are between 0.75 (sample 5 and 9) and 0.91 (sample 8). The structure of samples 6, 10, and 11 is affected by high shear forces and viscosity is diminished above 10 s^−1^, effect more pronounced for PL/curcumin system (sample 11). In the non-Newtonian region, the physical interactions between PL micelles continuously break and rebuild, and polysaccharide chains disentangle and tend to align along the shear field; thus, the overall dynamics are very complex, and it dictates the sample behavior.

From rheological investigations, it can be concluded that the viscoelastic characteristics and the flow behavior of PL/curcumin system can be improved by adding chitosan.

### 3.2. The Antimicrobial Behavior of the Pristine Polysaccharides and Pluronic F127-Based Gels

In the present study, the agar disc diffusion method, also known as the Kirby-Bauer method, was used to determine the antimicrobial activity of the samples. This method entails adding compounds to a solid culture medium that has already been pre-inoculated with a microbial suspension and the clear zone diameter, as a result of the growth inhibition around the film discs after 24 h at 37 °C, was measured. Data on the diameters of the inhibition zones of each tested sample are presented in Table 1. First of all, it can be noticed that only a few samples were distinguished by antimicrobial activity. Sample 1 containing chitosan presented a modest antimicrobial activity against almost all the tested reference strains (except the Gram-positive bacterial strain represented by *S. aureus*), without significant differences between species. As specified above, sample 0 (Pluronic F127 solution) and samples 2 up to 10 did not present antimicrobial activity. Among the investigated systems, only samples 11 and 12 demonstrated a high antimicrobial efficiency against all the tested bacterial and yeast strains. The antibacterial activity of these two samples was definitely higher than the antifungal activity. The addition of chitosan, which proved to be slightly efficient against almost all the tested samples, to the Pluronic and curcumin formula, determined the increase of sample 12 efficiency (as presented in Table 2 and Figure 8). In this particular case, it can be noticed that, as presented in Figure 8, the highest activity was registered against two Gram-negative bacterial strains represented by *E. coli* (up to 27 mm of inhibition zone) and especially against *P. aeruginosa* (up to 34 mm of inhibition zone).

Thermosensitive Pluronic F127 is a desirable biomaterial for many applications in regenerative medicine or dentistry [1,2,3,4,5,6,7,8]. By itself, this copolymer does not show antimicrobial activity and the observations from the present study (sample 0) confirm this hypothesis. According to literature data, Pluronics and Pluronic-lysozyme conjugate present bifunctionality, having both antiadhesive and antibacterial effects [54]. By hydrolyzing 1,4-beta-linkages between N-acetylglucosamine and N-acetylmuramic acid in the cell wall, lysozyme displays antibacterial activity against microorganisms, particularly Gram-positive bacteria. Additionally, modifying lysozyme conformation through physical or chemical interactions could result in an enhancement in the antibacterial activity of the lysozyme against Gram-negative bacteria [55]. Within this context and in the present study, none of the Pluronic F127 combinations with chitosan (sample 5), sodium alginate (sample 6), gellan gum (sample 7), k-carrageenan (sample 8), ZnO (sample 9) and lysozyme (sample 10) presented antimicrobial activity against none of the tested reference bacterial and yeast strains. Because of its biological origin, non-toxicity, hydrophilicity, biocompatibility, biodegradability, and low cost, sodium alginate has special qualities that make it ideal for use in a variety of biomedical applications. Several previous studies have focused on the usage of sodium alginate alone or in a composite matrix having antibacterial properties [56,57]. Though, as evidenced by the current investigation, sodium alginate does not exhibit antibacterial action on its own. A similar situation was noticed in the case of gellan gum. One glucuronic acid, one rhamnose, and two glucose sugars make up the tetrasaccharide repeats in gellan gum, a commercially significant exopolysaccharide from *Sphingomonas elodea* [58,59]. This compound possesses antimicrobial activity only in systems loaded with antibiotics, as previously presented in the literature [60,61]. Consequently, gellan gum did not display any antimicrobial activity following this study.

Due to its high hydrophilicity, mechanical strength, biocompatibility, and biodegradability, κ-carrageenan is primarily employed in the food area as a gelling, stabilizing, and thickening agent, but its antimicrobial activity could be achieved by incorporating nanoparticles or plant essential oils [62,63]. The fact that k-Carr did not present antimicrobial activity by itself or in combination with Pluronic F127 (samples 4 and 8, respectively) is in accordance with the literature.

According to several studies [64,65], the cationic nature of chitosan contributes to its antimicrobial activity, and the electrostatic interaction between positively charged sites and the negatively charged microbial cell membranes is what causes cellular lysis, this fact being assumed to be the primary antimicrobial mechanism [64,66]. As a result, it is anticipated that polymers with increased charge densities will have enhanced antibacterial activity. Data on the antibacterial activity of chitosan are widely presented in the literature, with results ranging from being quite effective against *S. aureus* [65,67,68], to slightly efficient against *E. coli* [68,69]. These results are partially confirmed by our study, and chitosan proved to have weak antimicrobial activity against all the tested samples, except the Gram-positive strain represented by *S. aureus*. It is interesting to note that some authors stated that the lack of uniform microbiological protocols while working with this natural biopolymer might be also a significant contributor to these differences [70]. This hypothesis should be taken into account for the other types of polymers tested in this study.

Turmeric (*Curcuma longa* L.) contains the primary bioactive compound curcumin, which has been shown to have potent antibacterial and antifungal properties against the most common microorganisms found in nososocomial infections [71,72,73,74]. As expected, the addition of curcumin into our systems leads to the achievement of antimicrobial activity. The presence of chitosan in the systems containing curcumin and Pluronic F127 increased the antimicrobial activity by synergism, achieving an inhibition zone that usually antibiotics have against all the tested samples. On the other side, the same formulation had a decreased antifungal activity.

## 4. Discussion

The temperature-induced gelation of PL solutions under physiological conditions can be explained by the structural characteristics of the amphiphilic copolymer having a PPO hydrophobic block flanked by two PEO hydrophilic blocks. By increasing polymer concentration and the temperature above well-defined critical values, micelles and polymicelles are formed [12] and suddenly evolve into a network structure. The rheological study can provide information on both sol and gel states [75]. The sol-gel transition determines a sharp increase of low shear rate viscosity and viscoelastic parameters. Due to its micellar structure, PL has poor structural integrity [16]. Another major deficiency of pristine PL gels is the lack of antibacterial activity.

Many studies were focused on natural compounds as promising alternatives to antibiotics for increasing the antimicrobial activity or cellular uptake of materials [51,76].

Polysaccharides are often used to improve rheological and biological characteristics. According to our knowledge, in the literature, there is no clear information about the simultaneous improvement of antimicrobial activity and gel properties induced by biomolecules introduced into Pluronic-based materials. In the present study, it was observed that the improvement of viscoelastic property by polysaccharide addition is not necessarily accompanied by a reasonable antimicrobial activity. Chitosan is considered an antimicrobial polysaccharide against filamentous fungi, yeasts, and bacteria (more active against Gram-positive as compared with Gram-negative bacteria) and a prebiotic that enhances colonization resistance against pathogens [70,77]. Low-molecular-weight chitosan chains present both extracellular and intracellular inherent antimicrobial activity [78,79]. Apart from such direct bactericidal and fungicidal effects, chitosan and its derivatives are efficient for treating enteric infections [77]. Chitosan is a potential candidate for many applications, for example, drug delivery carriers, flocking agents in water treatment, food additives, supplements for food preservation, dehydrating agents in cosmetics, components of hydrogel film in the pharmaceutical domain etc. [24,78].

However, pure chitosan as well as most of its derivatives present lower antimicrobial effects compared to clinical antimicrobial drugs. In addition, the efficiency of the antimicrobial action of chitosan depends on the type of targeted microorganism [78]. Moreover, in the present study, we observed that the antimicrobial activity of the PL/CS gels is not relevant for the PL samples containing other investigated biomolecules or ZnO nanoparticles. This important property was significantly enhanced for Pluronic F127 gel in the presence of curcumin, but the gelation is considerably delayed and flow behavior is altered at high shear rates. By including curcumin and chitosan in Pluronic gels, in the presence of acetic acid that favors chitosan and also curcumin dissolution, both viscoelastic and antimicrobial properties were considerably improved. Stored at 37 °C for at least 90 min, PL/CS/curcumin gels present good stability, solid-like properties (G′ > G″, tanδ < 1), improved gel strength and stability as compared with PL gels, being suitable candidates for topical applications.

Pluronic F127 macromolecules dissolved in an aqueous medium only comply with some criteria for an ideal injectable gel: low viscosity in sol state (a fluid-like behavior below the gelation temperature, with Newtonian viscosity depending on temperature and concentration [80]); fast sol-gel transition as the temperature increases; in a gel state, yield stress and shear-thinning behavior (above the sol-gel transition temperature, the sample possesses yield stress and presents high viscosity at a low shear rate and shear thinning at high shear rates) [80,81]. In the present paper, the viscosity decreases with 4 orders of magnitudes from 0.01 s^−1^ to 100 s^−1^, with good printing properties into complex structures, with high fidelity [8]. However, efforts are being made to fulfill the most important missing characteristics, which are poor biocompatibility (long-term culture of encapsulated cells, cell adhesion ability) and lack of enzymatic degradation [4,82,83]. The modest performances concerning cell viability were attributed to structural disruption and rapid dissolution of the hydrogel; thus, there is an incapacity of cells to be connected to this network [4,84]. Moreover, due to the micellar structure of Pluronics, the hydrogels erode over time and lose their ability to maintain structural integrity [17], the networks become soluble in an aqueous environment, and after one week, they are readily dissolved [82].

Pluronic F127, one of the most important thermoreversible gels, was regarded as nontoxic and it was applied in localized drug delivery, such as intramuscular [85], intraperitoneal [86], and subcutaneous [87] injections. However, PF127 alone is not considered as the optimal formulation for drug delivery due to the low dissolution of hydrophobic drugs, short duration in the subcutaneous layer, usually less than 3 days, and non-biodegradability [88]. The cytotoxicity data indicated that the toxic potential of Pluronic may be related to the unimers and not to the micelles. There are different biological effects of hydrophobic Pluronic unimers, which are more toxic as compared with their corresponding micelles [89,90].

Derivatization or modification of copolymer structure was carried out to alleviate some of their shortages [90,91]. These supplementary steps for preparing the gel are time-consuming, and they occasionally introduce a higher level of toxicity into the sample. The addition of polysaccharides and proteins to Pluronics gels for injectable formulations represents a promising non-toxic approach [7,81]. Pluronic F127 gels incorporating gelatin and curcumin promote the adhesion and proliferation of fibroblast cells; they improve the burn-healing process and reduce scar formation [92]. Mixtures of 18% and 20% (*w*/*w*) Pluronic F127 and 4% *w*/*w* methylcellulose have shown shear-thinning and thixotropic behavior, printability, improved mechanical properties, and good cell viability [93]. The cytotoxicity studies of Pluronic F127-based formulations showed cell viability higher than 80% for human HaCaT keratinocytes for concentrations lower than 20 μg/mL [22]. Low cytotoxicity (cell viability > 85 %) on Bel 7402 and L02 cell was reported for α-tocopherol modified Pluronic F127 micelles which interact via redox-sensitive disulfide bonds (concentrations ranging from 12.5 μg/mL to 200 μg/mL) [94]. After 48 h of incubation, cell viability greater than 85% was reported for composite hydrogels consisting of curcumin-loaded F127 micelles and 5-fluorouracil dispersed into chitosan/oxidized dextran networks [95].

Overall, all fundamental studies on Pluronic-based systems provide additional research opportunities, increasing the potential of suitable formulations for clinical applications.

## 5. Conclusions

In the present paper, the viscoelastic properties and antimicrobial activity of Pluronic gels in the presence of different biomolecules or ZnO nanoparticles were tested, as schematically shown in Figure 9.

It was demonstrated that some polysaccharides, proteins or nanoparticles do not have antimicrobial activity. Chitosan dissolved in the presence of acetic acid is an exception, but its antimicrobial activity is moderate and it does not induce antimicrobial activity in Pluronic gels. However, the polysaccharide addition can bring other benefits and here we exemplify the viscoelastic behavior in various shear conditions, which is also temperature dependent. On the other hand, curcumin has a strong antimicrobial effect.

The organic solvents (in our study acetic acid) added to water facilitate the curcumin and chitosan dissolution, but, according to the present rheological study, they change the gelation kinetics. In addition, curcumin interacts with the hydrophobic core of Pluronic and delays the gelation. Chitosan and curcumin present a synergistic effect, and their presence gives stability in time and fluidity during the injection. The main conclusion of this study is that an optimum formulation suitable for an injectable composite gel represents a compromise between viscoelastic characteristics and antimicrobial activity.

## Figures and Tables

**Figure 1 polymers-15-00355-f001:**
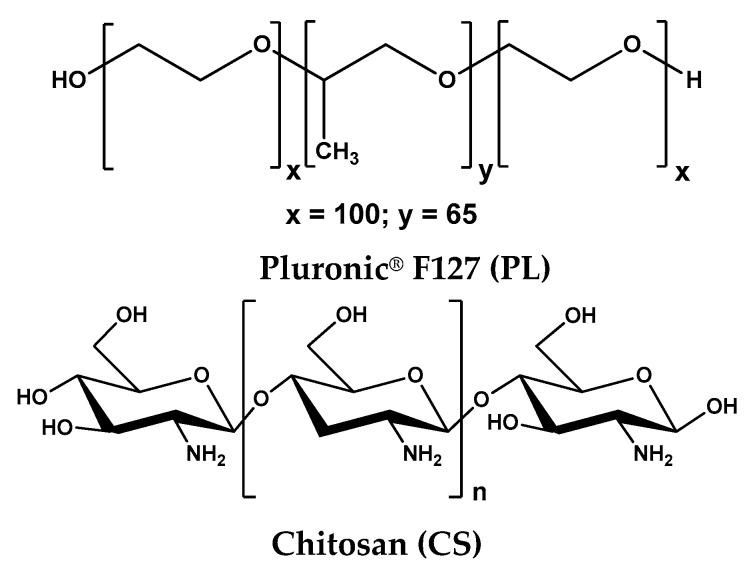
The chemical structures of individual components used for gel preparation.

**Figure 2 polymers-15-00355-f002:**
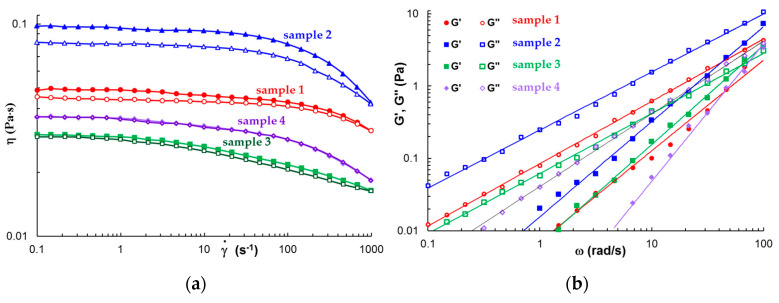
The rheological behavior of 1% polysaccharide solutions at 37 °C: (**a**) Shear viscosity at increasing (full symbol) and decreasing (open symbol) shear rate (steady shear conditions). In order to avoid the overlapping of the curves, the data for sample 4 were shifted down by a factor of 1.35. (**b**) G′, G″ as a function of ω (γ = 0.1%).

**Figure 3 polymers-15-00355-f003:**
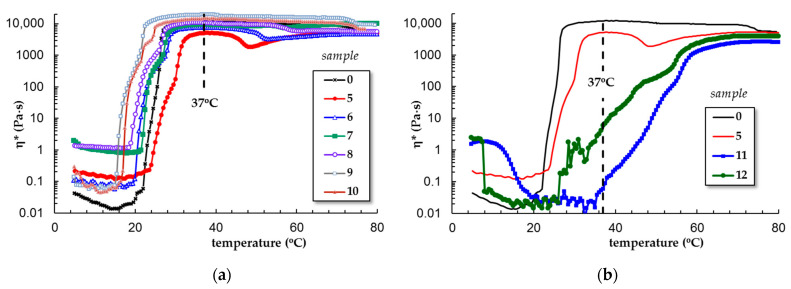
Variation of the complex viscosity with temperature for PL-based gels: (**a**) in aqueous solution; (**b**) in solution of 0.33% acetic acid (heating rate of 1 °C/min, ω = 1 rad/s, γ = 1%).

**Figure 4 polymers-15-00355-f004:**
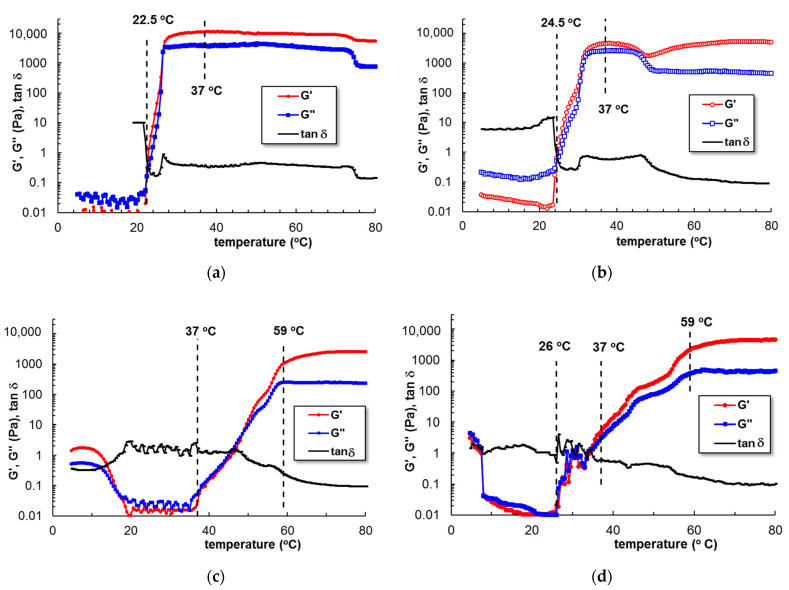
G′, G″ and tanδ as a function of temperature for (**a**) sample 0; (**b**) sample 5; (**c**) sample 11; (**d**) sample 12 (heating rate of 1 °C/min, ω = 1 rad/s, γ = 1%). All samples contain 0.33% acetic acid.

**Figure 5 polymers-15-00355-f005:**
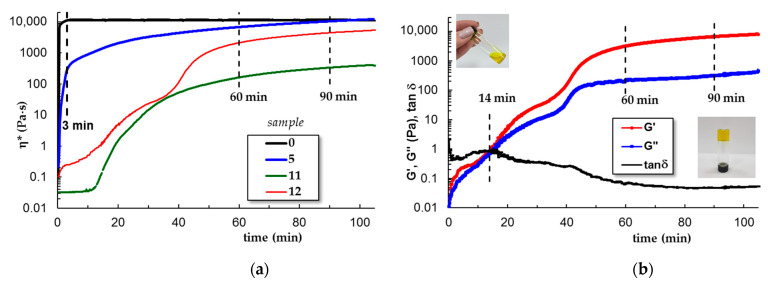
The kinetics of gelation at 37 °C (ω = 1 rad/s and γ = 1%): (**a**) complex viscosity as a function of time for samples 0, 5, 11 and 12; (**b**) G′, G″ and tanδ as a function of time for sample 12. The samples stored at 5 °C were poured on the lower plate of the rheometer and the temperature was set at 37 °C at the beginning of the measurement. All samples contain 0.33% acetic acid.

**Figure 6 polymers-15-00355-f006:**
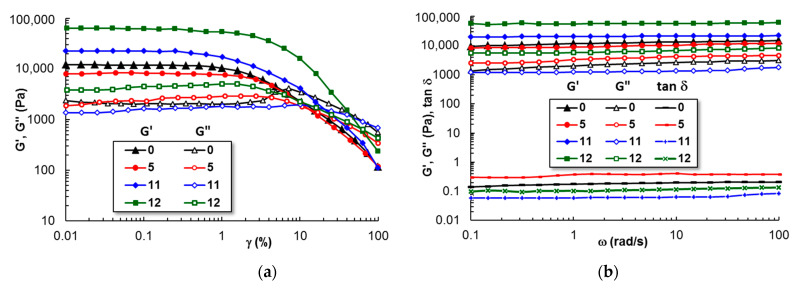
The viscoelastic parameters in (**a**) amplitude sweep and (**b**) frequency sweep tests for the PL-based gels. The samples were stored at 37 °C for 24 h.

**Figure 7 polymers-15-00355-f007:**
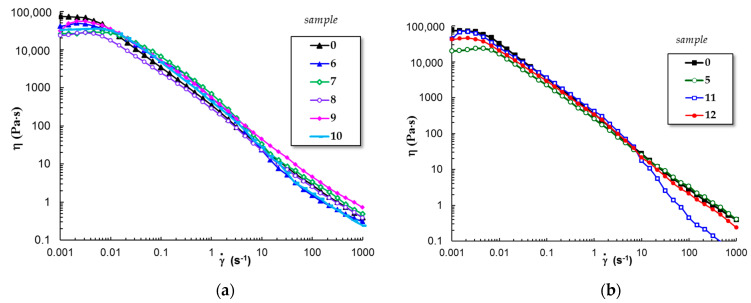
Shear flow behavior of PL based gels (**a**) in aqueous solutions and (**b**) in presence of 0.33% acetic acid. Before the flow test, the samples were stored at 37 °C for 24 h.

**Figure 8 polymers-15-00355-f008:**
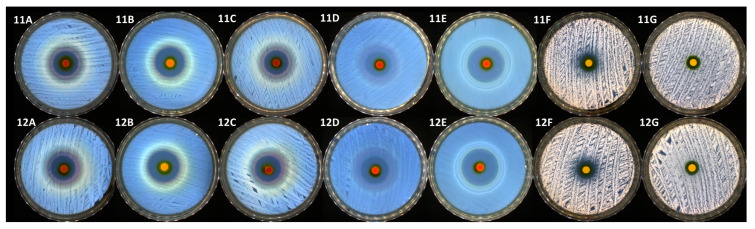
Antimicrobial activity of samples 11 and 12 against (**A**) *S. aureus*; (**B**) *E. coli*; (**C**) *E. faecalis*; (**D**) *P. aeruginosa*; (**E**) *K. pneumoniae*; (**F**) *C. albicans*; (**G**) *C. glabrata*.

**Figure 9 polymers-15-00355-f009:**
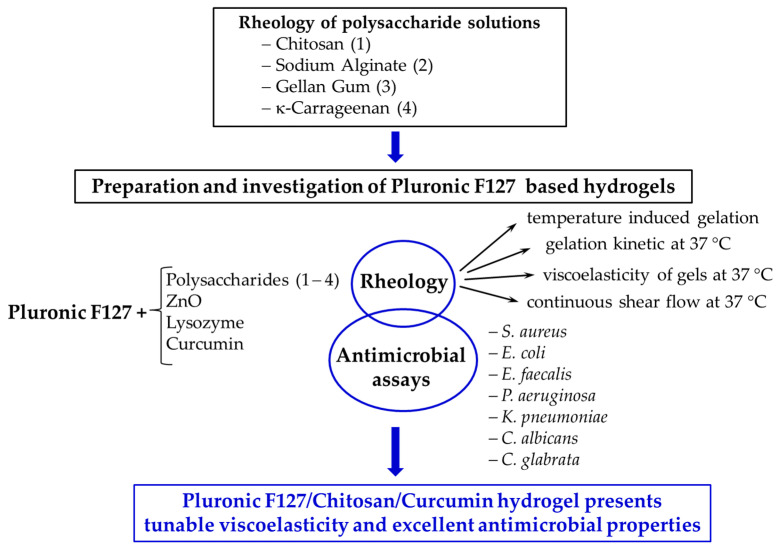
Brief presentation of the selected systems and the main investigations of this study.

**Table 1 polymers-15-00355-t001:** The composition of the samples and some rheological characteristics at 37 °C.

Sample Code	Sample Composition	G′ (Pa) (10 rad/s)	G″ (Pa) (10 rad/s)	η_o_ (Shear) (Pa·s)
0 *	Pluronic^®^ F127 (PL)	13,200	2600	74,000
1 **	Chitosan (CS)	0.0254	0.0622	0.0488
2 **	Sodium Alginate (Alg)	0.3067	1.5612	0.0976
3 **	Gellan Gum (GG)	0.1718	0.2914	0.0318
4 **	k-Carrageenan (κ-Carr)	0.1554	0.5511	0.0446
5 *	PL/CS	8800	3300	23,300
6 *	PL/Alg	8990	2450	47,400
7 *	PL/Gell	8680	2310	28,600
8 *	PL/k-Carr	10,300	2990	27,900
9 *	PL/ZnO	19,500	3670	54,600
10 *	PL/Lysozyme	42,500	5850	35,200
11 *	PL/Curcumin	21,100	1320	61,500
12 *	PL/CS/Curcumin	57,583	6733	45,500

* total polymer concentration is of 16.83% wt.; ** polymer concentration is of 1% wt.

**Table 2 polymers-15-00355-t002:** Antimicrobial activity of the tested samples against the reference strains.

Sample Code	Inhibition Zone (mm)						
	*S. aureus*	*E. coli*	*E. faecalis*	*K. pneumoniae*	*P. aeruginosa*	*C. albicans*	*C. glabrata*
0	-*	-*	-*	-*	-*	-*	-*
1	-*	8.45 ± 0.35	8.40 ± 0.14	8.30 ± 0.14	8.80 ± 0.14	7.90 ± 0.14	7.50 ± 0.42
2 to 10	-*	-*	-*	-*	-*	-*	-*
11	21.35 ± 0.63	25.00 ± 0.14	21.30 ± 0.14	21.00 ± 0.14	33.30 ± 0.14	14.80 ± 0.57	11.60 ± 0.99
12	22.60 ± 0.28	26.95 ± 0.21	25.40 ± 2.68	22.25 ± 0.07	34.15 ± 0.91	15.30 ± 0.56	12.60 ± 0.28

-* these samples did not present antimicrobial activity against the tested reference strains.

## Data Availability

Data are available on request.

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
