# Peer review of "Temperature Induced Gelation and Antimicrobial Properties of Pluronic F127 Based Systems"

_polymers, 2023, doi:10.3390/polym15020355_

Round 1

Reviewer 1 Report

Dear Authors,

I studied your manuscript entitled "Viscoelastic and antimicrobial properties of Pluronic F127 gels". This paper comprises interesting results that certainly deserve publication. Some spaces need to be improved in terms of journal quality. I recommend a minor revision before further consideration for publication in the Polymers.

1) The quality of the abstract and conclusion should be enhanced by the inclusion of significant research findings. More quantitative information in these sections would be beneficial.

2) This manuscript has a phenomenological style, observing a result and explaining it with statements. It would be helpful if you conducted more analysis based on published research.

3) Page 6, paragraph 1: You should present the curve of the oscillatory shear experiment (frequency sweep test).

4) The research question needs to be more well-stated and discussed. It would also be beneficial if the manuscript included a brief comparison of the antibacterial properties of gels and nanofibrous scaffolds. Some related papers were strongly suggested to be used:

a) https://doi.org/10.3390/polym14091637

b) https://doi.org/10.3389/fbioe.2022.1027351

c) https://doi.org/10.3390/pharmaceutics14061208

d) https://doi.org/10.1021/acsami.2c04849

5) You can present more analyses that evaluate the properties of the prepared gels. Therefore, the results of additional analyses, such as in vitro degradation, MTT, cell adhesion assay, etc., could be presented and discussed.

6) English language needs some polishing since some terms are vague. The paper's title is also recommended to be revised.

Author Response

Comments and Suggestions for Authors

Dear Authors,

I studied your manuscript entitled "Viscoelastic and antimicrobial properties of Pluronic F127 gels". This paper comprises interesting results that certainly deserve publication. Some spaces need to be improved in terms of journal quality. I recommend a minor revision before further consideration for publication in the Polymers.

1) The quality of the abstract and conclusion should be enhanced by the inclusion of significant research findings. More quantitative information in these sections would be beneficial.

In situ gelling systems as aqueous polymeric solutions characterized by a low viscosity before administration and, once administered, form a gel at the application site, able to control the release of the loaded drug. The sol–gel transition depends on one or more stimuli, which can be exogenous as ultraviolet irradiation or endogenous like temperature variation, pH change, ionic cross-link formation, and solvent exchange. The advantages of these formulations encompass ease of application and prolonged permanence at the administration site, as well as protection of the drug from environmental conditions, modulation of drug release and, depending on the polymer used and the application site.

We thank the reviewer for the careful analysis of the manuscript and for his comments. We did our best to improve the whole manuscript, including the abstract and conclusion.

2) This manuscript has a phenomenological style, observing a result and explaining it with statements. It would be helpful if you conducted more analysis based on published research.

Based on published data, we improved the comments in the revised manuscript.

3) Page 6, paragraph 1: You should present the curve of the oscillatory shear experiment (frequency sweep test).

The results obtained for polysaccharide solutions in oscillatory shear experiments were included in Figure 2b.

4) The research question needs to be more well-stated and discussed. It would also be beneficial if the manuscript included a brief comparison of the antibacterial properties of gels and nanofibrous scaffolds. Some related papers were strongly suggested to be used:

  1. a)https://doi.org/10.3390/polym14091637
  2. b) https://doi.org/10.3389/fbioe.2022.1027351
  3. c) https://doi.org/10.3390/pharmaceutics14061208
  4. d) https://doi.org/10.1021/acsami.2c04849

The suggested references were analysed in the context of the present investigation.

The following sentences were added in Introduction:

<< On the other hand, the proteins and renewable polysaccharides are frequently used for the preparation of biomaterials in order to improve their biodegradability, biocompatibility and mucoadhesive behavior. Antimicrobial activity always represents a challenge for the design of new biomaterials because in some cases the enhancement of physico-chemical performances implies a worsening of the biological properties or a weakening of the network structure. Various polymers and biologically active compounds were used to develop suitable platforms for tissue engineering and regenerative medicine. As for example, electrospun surgical sutures with antibacterial properties should promote a fast wound healing [9].The parameters selected for the electrospinning process or the use of various organic solvents can affect the biological activity of the medical sutures [9,10]. Various approaches were proposed to prevent bacterial growth. Simultaneous incorporation of different antimicrobial agents, such as curcumin and zinc oxide nanoparticles into the electrospun chitosan/ polycaprolactone [11] or inorganic nanoparticles and natural essential oils loaded into Janus fibers [12] inhibited bacterial growth in the case of Escherichia coli and Staphylococcus aureus. The use of coaxial fibers as mesh for guided tissues regeneration allowed their functionalization with bioactive compounds. These antibacterial coaxial membranes can be long-term preserved and the drug (rifampicin) efficiently released when required [13]. >>

In addition, the Introduction section was improved with similar data reported in the literature concerning antimicrobial Pluronic gels.

5) You can present more analyses that evaluate the properties of the prepared gels. Therefore, the results of additional analyses, such as in vitro degradation, MTT, cell adhesion assay, etc., could be presented and discussed.

We are aware that additional analyses are required, but at this point the main goal of this paper was to present the viscoelastic and antimicrobial properties of Pluronic F127 gels in combination with several types of polysaccharides (and not only) and to select a suitable system with adequate characteristics for an injectable gel. The suggested analyses will be taken into account for an additional study which will which will be accomplished to achieve results in terms of biocompatibility and cytotoxicity.

6) English language needs some polishing since some terms are vague. The paper's title is also recommended to be revised.

English was carefully revised and the title of the paper was changed as:

Temperature induced gelation and antimicrobial properties of Pluronic F127 based systems

We thank the reviewer for the careful review of the manuscript and helpful suggestions.

Reviewer 2 Report

This manuscript by Lupu et al. reports the viscoelastic and antimicrobial properties of Pluronic F127 gels. I recommend a minor revision.

1. I suggest authors to add a scheme describing the main content of this study at the beginning of the manuscript.

2. The morphology of the gel should be characterized by TEM.

3. The INTRODUCTION section should be enriched by adding more systems for antimicrobial especially the gels.

Author Response

Comments and Suggestions for Authors

This manuscript by Lupu et al. reports the viscoelastic and antimicrobial properties of Pluronic F127 gels. I recommend a minor revision.

  1. I suggest authors to add a scheme describing the main content of this study at the beginning of the manuscript.

We thank the reviewer for this suggestion. A scheme describing the main content of the study was inserted at the end of the manuscript.

Figure 9. The selected systems and the main investigations of this study

  1. The morphology of the gel should be characterized by TEM.

We thank the reviewer for this suggestion. The present investigation was carried out to explore the viscoelastic properties of Pluronic based gels in presence of polysaccharides and we tried to find suitable system in order to enhance the antimicrobial activity. For the next step, we intend to test other polysaccharides (such as xanthan gum) and to do a morphological study of the hydrogels in correlation with their viscoelastic properties.

In the revised manuscript, we introduced some aspects discussed in literature concerning the morphology of Pluronic gels.

  1. The INTRODUCTION section should be enriched by adding more systems for antimicrobial especially the gels.

The Introduction section was improved with new data from literature concerning antimicrobial Pluronic gels.

We thank the reviewer for the careful analysis of the manuscript and for making useful suggestions.

Round 2

Reviewer 1 Report

Dear Authors,

Thank you for considering my comments and for the deep revision which enhances both the clarity and the relevance of your work. I have recommended the publication of your article as is.

Author Response

Dear Reviewer and Editors,

We took into consideration your recommendation and we added a supplementary comment at the end of the Discussion section.

A couple of references from the revision 1 have been discarded, but we have added new ones in the added paragraph.

We thank reviewers and editors for the careful analysis of the manuscript and their insightful comments.

We hope that after revision, our manuscript has been considerably improved and meets the demands of the journal.

At the beginning of 2023, we wish you good health, joy and many achievements.

Yours sincerely,

Maria Bercea and coworkers
